# Long-Term Follow-Up of Device-Assisted Clampless Off-Pump Coronary Artery Bypass Grafting Compared with Conventional On-Pump Technique

**DOI:** 10.3390/ijerph19010275

**Published:** 2021-12-27

**Authors:** Carlo Bassano, Paolo Nardi, Dario Buioni, Laura Asta, Calogera Pisano, Fabio Bertoldo, Claudia Altieri, Giovanni Ruvolo

**Affiliations:** Division of Cardiac Surgery, Tor Vergata University Hospital, Viale Oxford 81, 00133 Rome, Italy; carlo.bassano@uniroma2.it (C.B.); docyuk@libero.it (D.B.); laura.asta@uniroma2.it (L.A.); calogera.pisano@uniroma2.it (C.P.); fabio.bertoldo@uniroma2.it (F.B.); claudia.altieri@ptvonline.it (C.A.); giovanni.ruvolo@uniroma2.it (G.R.)

**Keywords:** coronary artery bypass grafting, off-pump CABG, device-assisted proximal venous graft anastomoses

## Abstract

Study objective: To evaluate the long-term outcomes of clampless off-pump coronary artery bypass grafting (C-OPCAB) compared with conventional on-pump double clamping coronary artery bypass grafting (C-CABG). Methods: From October 2006 to December 2011, 366 patients underwent isolated coronary artery bypass grafting. After propensity score matching of preoperative variables, 143 pairs were selected who received C-OPCAB with the use of device-assisted PAS-Port proximal venous graft anastomoses or C-CABG, performed by the same surgeon experienced in both techniques. Data of the two groups of patients were retrospectively analyzed up to 14 years of follow-up. Results: As compared with C-OPCAB, in the C-CABG patients, the performed number of grafts per patient was higher (2.9 ± 0.5 vs. 2.6 ± 0.6, *p*-value 0.0001). At 14 years, overall survival, including in-hospital death, was 64 ± 4.7% for the C-OPCAB vs. 55 ± 5.5% for the C-CABG, freedom from overall MACCEs 51 ± 6.2% vs. 41 ± 7.7%, and from late cardiac death 94 ± 2.4% vs. 96 ± 2.2% (*p*-value not significant, for all comparisons). No significant statistical differences were observed in the actual rates of adverse events during follow-up. Independent predictors of survival were advanced age at operation (*p*-value 0.001) and a lower mean value of preoperative left ventricular ejection fraction (*p*-value 0.015). Conclusions: Our single-center study analysis suggests that clampless OPCAB using device-assisted proximal anastomoses proved to be not inferior to double-clamping CABG in the long-term follow-up, provided that involved surgeons are familiar with both techniques. These conclusions are supported by a large and long-term follow-up period, eliminating potential bias, i.e., by means of the propensity score matching and analyzing single-surgeon experience.

## 1. Introduction

On-pump coronary artery bypass grafting (CABG) is considered the gold-standard treatment for severe CAD [1,2]. The off-pump procedure (OPCAB), which was assumed to be less harmful since it avoids CPB, actually failed to show significant early advantages over CABG, while long-term results were often inferior to those achieved with the conventional technique [3,4].

Nonetheless, in very fragile patients, or in cases with the severely diseased ascending aorta, OPCAB proved to be useful in preventing neurological or aortic complications, provided that any clamping was prevented (anaortic or clampless OPCAB, i.e., C-OPCAB) [5]. Facilitating devices have been developed to abstain from aortic side-clamping when proximal graft suturing was required. Among those, The Cardica PAS-Port automatic system device (Cardica Inc., Redwood City, CA, USA) allows proximal, automatic, one-step anastomosis of the saphenous vein graft without aortic clamping. There have been early concerns about the device’s performance, but it eventually proved to be highly reliable with an excellent early and mid-term patency rate and useful in preventing possible aortic particulate embolism [6,7].

However, the device’s long-term results have not yet been reported. The aim of this study is to compare long-term outcomes, i.e., survival and freedom from major adverse cardiac and cerebrovascular events (MACCEs) of patients submitted to either on-pump CABG with double clamping (i.e., C-CABG) or clampless OPCAB with total arterial revascularization or PAS-port facilitated proximal anastomoses (i.e., C-OPCAB).

## 2. Materials and Methods

### 2.1. Study Population

From October 2006 to December 2011, 366 patients affected by multivessel coronary artery disease underwent surgical myocardial revascularization in our division performed by a single senior surgeon, experienced in both techniques: 146 patients underwent C-OPCAB, 223 patients underwent C-CABG. The study was approved by our Local Institutional Review Board. All patients gave their informed surgical consent.

The choice of selecting a single-surgeon cohort was dictated by the need to eliminate the potential bias related to surgeon experience. Moreover, since the patients were not randomized to a treatment group, a propensity score matching was performed to avoid selection bias related to preoperative characteristics of the patients. The propensity score was performed by applying the logistic regression, including the following preoperative variables: age, sex, left ventricular ejection fraction, creatinine serum level, hypertension, smoke habit, diabetes mellitus, previous stroke or transient ischemic attack, carotid artery stenosis >50% <70%, left main coronary disease >50%, previous myocardial infarction, percutaneous coronary revascularization, carotid artery stenting, and urgent operation. After propensity score adjustment, 143 pairs were selected, i.e., 143 C-OPCAB: 143 C-CABG. The characteristics of the unmatched and matched patient population are reported in Table 1. Patients requiring coronary surgery reoperation or concomitant procedures, i.e., valve surgery, replacement of ascending aorta, and ventricular resection, were excluded from the study.

### 2.2. Criteria to Choose C-OPCAB and C-CABG

The choice of performing the on-pump conventional CABG as the only treatment was based on the left ventricular systolic function. Conventional on-pump CABG was performed in the presence of left ventricular ejection fraction less than 30%, left ventricular end-diastolic diameter greater than 60 mm, distal diffuse narrowing of coronary arteries, intramyocardial course of the left anterior descending coronary artery, surgery required in the presence of perioperative hemodynamic instability, and availability of the devices required for off-pump surgery.

### 2.3. Surgical Strategy and Patients’ Mangement

A complete longitudinal sternotomy was performed in all patients. C-CABG was performed by means of normothermic cardiopulmonary bypass and intermittent antegrade warm blood cardioplegia (600 mL the first dose, 400 mL the others administered every 20–25 min). Cardiopulmonary bypass was performed by the use of a Sorin Monolyth-Pro (Sorin Biomedica; Turin, Italy) or Capiox (Terumo Cardiovascular System; Borken, Germany) membrane oxygenator and a Stockert roller pump (Stockert Instrumente; Munich, Germany). C-OPCAB was performed with an all-arterial grafts approach in 36 patients (i.e., totally anaortic), or by means of automated proximal anastomoses of the venous grafts with the Cardica PAS-Port (Cardica; PAS-Port Proximal Anastomosis System, Redwood City, CA, USA). PAS-port proximal venous graft anastomoses were performed in 107 patients (i.e., partially anaortic); 115 were PAS-port proximal anastomoses; 159 were PAS-port dependent distal anastomoses, i.e., in the presence of multiple sequential venous grafts. The stabilization of the heart was obtained with the use of Octopus and Starfish (Medtronic Inc., Minneapolis, MN, USA) or with the use of Acrobat and X-pose (Guidant Co.; Boston Scientific, Boston, MA, USA) later on. To perform C-OPCAB, in all cases, it was required perfusionist’s stand-by on a ready-dry state (i.e., mounted, non-primed cardiopulmonary bypass circuit). The continuous perfusion of the coronary arteries was maintained after arteriotomy by means of intravascular shunts (Clearview, Medtronic Inc., Minneapolis, MN, USA). Monitoring of cardiac function was obtained with trans-esophageal echocardiography and insertion of a Swan-Ganz pulmonary artery catheter. Internal thoracic artery, as in situ graft, pedunculated or skeletonized, was used as a conduit of choice for the revascularization of the left anterior descending artery.

### 2.4. Data Collection

Perioperative myocardial infarction was defined as an increase of postoperative troponine I higher than 5 ng/mL associated with a Creatine PhosphoKinase-MB above normal values and more than 10% of total Creatine PhosphoKinase value, and the onset of ECG new anomalies. Pulmonary complication was defined as primary lung failure requiring mechanical ventilation for more than 48 h, reintubation, or intermittent application of positive end-expiratory pressure by mask. Neurologic injury was defined as non-reversible, i.e., lethal coma or stroke, or reversible, i.e., transient ischemic attack or postoperative delirium as described in the Diagnostic and Statistical Manual of Mental Disorders requiring prolonged mechanical ventilation and/or ICU and in-hospital stay. Renal insufficiency was defined as a 2-fold increase of preoperative serum creatinine level or oliguria necessitating continuous veno-venous filtration. Operative mortality included death in hospital after operation at any time or within 30 days after discharge. The major adverse cardiac and cardiovascular events (MACCE) endpoint was defined as the composite incidence of all-cause death, documented myocardial infarction, repeat coronary revascularization of the target lesion, return of low-threshold angina, and stroke.

The clinical status of every patient was ascertained during a 2-month period (April–May 2021). All patients had at least 60 months of follow-up; the mean duration of follow-up was similar in the C-OPCAB vs. C-CABG (92 ± 50 months vs. 90 ± 44 months), reaching 14 years; the clinical follow-up included 1770 patients-year and was 85% complete. All causes of operative and at follow-up death; need for in-hospital readmission for cardiovascular causes, and functional status of the patients were also recorded at the outpatient clinic visit or by telephone interview. Collected data of functional tests and echocardiographic examinations during the follow-up were analyzed.

### 2.5. Statistical Analysis

Analysis was performed with Stat View 4.5 (SAS Institute Inc., Abacus Concepts, Berkeley, CA, USA). The rates of postoperative events were compared with *χ^2^* contingency tables and Fisher’s exact test for categorical variables and with the student’s *t*-test for continuous unpaired variables. Risk factors analysis to detect independent predictor/s for postoperative outcomes, i.e., mortality or neurological injury, was performed using the Logistic Regression analysis. Overall survival, including in-hospital deaths, freedom from MACCEs, and from late cardiac death, were expressed as mean values plus or minus 1 SD and computed by using the Kaplan–Meier method. The log-rank test was used to compare survival estimates among subgroups, i.e., C-OPCAB vs. C-CABG, and the Cox proportional hazards methods were used to evaluate the influence of variables on time as predictors of death and adverse events in the entire patient population. All continuous values were expressed as mean plus or minus 1 SD of the mean. A *p*-value less than 0.05 was considered statistically significant.

## 3. Results

### 3.1. In-Hospital Outcomes

The number of grafts per-patient performed was 2.6 ± 0.6 in the C-OPCABG group and 2.9 ± 0.5 in the C-CABG group (*p*-value 0.0001). Complete revascularization was achieved in 97.2% of the patients in the C-OPCAB group and in 98.6% in the C-CABG group, respectively (*p*-value not significant).

Operative mortality was 3.1% (*n* = 9); 1.4% in the C-OPCAB group, and 4.9% in the C-CABG group, respectively. Five deaths were due to cardiac causes. Four patients among the seven deaths in the C-CABG group presented preoperatively an EuroSCORE II > 8. In particular, operative mortality was higher in patients undergoing surgery under urgency/emergency settings due to evolving myocardial infarction and/or hemodynamic instability (*n* = 8/86). Independent predictors of operative mortality were the preoperative lower value of LVEF (*p*-value 0.001; Hazard Ratio [HR], 3.2), urgency (*p*-value 0.01; HR, 2.5), evolving myocardial infarction (*p*-value 0.03; HR, 2.1), previous stroke (*p*-value 0.03; HR, 2.2). The global incidence of neurological injury was 2.1% (*n* = 3) in the C-OPCAB group and 9.1% (*n* = 11) in the C-CABG group (*p*-value 0.02). As compared with the C-OPCABG group, the higher incidence of neurological injury observed in the C-CABG group was due primarily to a greater number of reversible neurological events, i.e., transient ischemia or delirium (6.3% or 9 patients vs. 2.1% or 3 patients). Two patients in the C-CABG group experienced a postoperatively stroke (1.4% or 2 patients).

At the Logistic Regression analysis, off-pump CAB was found to be a protective factor against operative mortality (*p*-value 0.01; HR, −2.4) and the onset of postoperative neurological injury with a slight level of significance at the Regression analysis (*p*-value 0.04; HR, −2.0).

The rates of perioperative myocardial infarction, low cardiac output syndrome, stroke, acute kidney injury, pulmonary complications, and the stay in the intensive care unit were similar in both groups.

### 3.2. Follow-Up Results

At 14 years, overall actuarial survival, including in-hospital death, was 64 ± 4.7% for the C-OPCABG vs. 55 ± 5.5% for the C-CABG (*p*-value not significant) (Figure 1), freedom from MACCEs 51 ± 6.2% vs. 41 ± 7.7% (*p*-value not significant) (Figure 2), and from late cardiac death 94 ± 2.4% vs. 96 ± 2.2% (*p*-value not significant) (Figure 3), respectively. No significant statistical differences were observed in the actual rates of adverse events during the follow-up (Table 2). At the multivariate Cox Regression analysis, independent predictors of survival were advanced the mean age at operation (*p*-value < 0.001) and the lower mean value of preoperative LVEF (*p*-value 0.015). The only independent predictor of late cardiac mortality remained the lower mean value of preoperative LVEF (*p*-value < 0.001) (Table 3). At the Log-rank Mantel-Cox test, 14-year actuarial survival stratified for the preoperative LVEF value was 63 ± 4.2% vs. 29 ± 9.9% for LVEF > 0.35 and ≤0.35, respectively (*p*-value < 0.0001).

## 4. Discussion

Surgical myocardial revascularization performed on cardiopulmonary bypass and cardioplegic myocardial protection is considered the gold standard for the treatment of patients affected by multivessel coronary artery disease since it provides a motionless, bloodless field for optimal construction of distal coronary anastomoses. The ongoing technological advancements led to improved outcomes of CABG even in patients affected by several comorbidities. The execution of an off-pump surgical procedure was believed to assure better early results in the subset of frail patients, but, while immediate results substantially failed to show any obvious early advantage over conventional CABG [8,9,10], the long-term outcome appeared to be often inferior [11,12,13], probably due to less complete revascularization and/or less accurate suturing technique [14,15,16,17,18,19]. Difficulty in target vessel exposure, and mechanical or electrical instability during heart manipulation and positioning are assumed to be the main causes of decreased technical accuracy or induction to give up a difficult anastomosis [20]. Subsequently, disappointing long-term results, as decreased survival and ischemic recurrence, are easily expectable unless the attending surgeon is trained enough to afford critical situations with no detrimental effects on the procedure [21,22,23]. Therefore, only experienced surgeons are considered to be justified in performing OPCAB [24].

The reported rate of perioperative stroke in several series of patients undergoing C-CABG is not negligible, ranging from 1.1% to 4.6% [21,23,24]. In fact, surgical handling of the severely atherosclerotic or calcified ascending aorta during cannulation and cross- and side-clamping can increase the risk of serious complications, such as particulate embolism and neurological damage or aortic injury [25]. In this peculiar setting, the use of OPCAB regains relevance, as long as aortic manipulation is completely avoided also for clamping (anaortic OPCAB, A-OPCAB). This is usually achieved with the use of “total arterial” revascularization. When a proximal graft anastomosis is required on a diseased proximal aorta, the use of several devices that avoid any side-clamping is considered helpful, leading to results comparable to those achieved with totally anaortic clampless OPCAB. In the C-OPCAB group, we did not observe any cases of perioperative stroke.

The aim of this study is to verify if the early advantages provided by the use of an automatic saphenous graft connector, already demonstrated in the early and mid-term period following C-OPCAB [26,27], were maintained at a long-time follow-up.

The secondary endpoint was to determine if C-OPCAB performed by an experienced surgeon has long-term results similar to those achieved with C-CABG.

In the literature, some concerns exist on whether off-pump CABG is associated with inferior long-term outcomes. In the CORONARY trial, there was no difference in the composite outcome of death, myocardial infarction, stroke, and renal failure at 5 years of follow-up between off- and on-pump CABG (23.1% vs. 23.6%) [28]. On the contrary, in the ROOBY trial, 5-year mortality was significantly worse in the off-pump group (15.2% vs. 11.9%) [29].

In a single-center observational study of 12 812 patients from Emory University, Atlanta, GA, USA, there was no difference in 10-year mortality between on- and off-pump surgery after propensity score covariate adjustment [30]; of note, the authors reported that the key to long-term survival was the completeness of revascularization in both on- and off-pump patients. In a study from the United Kingdom performed on more than 13,000 propensity-matched patients followed for 13 years, there was no difference in survival, suggesting that when OPCAB is performed by highly experienced surgeons, there is no adverse effect on survival [31]. In contrast, a propensity-matched single-institution study from the Baylor Research Institute showed an elevated risk of late mortality at 10 years with OPCAB (HR: 1.18; 95% CI, 1.02–1.38) [19]. Finally, the most recent meta-analysis, including RCTs with >4-year outcomes on 8145 patients, reported an odds ratio for long-term mortality of 1.16 for OPCAB [32].

The evaluation of a single-surgeon cohort in our study was used to avoid biases related to surgical experience. Furthermore, the propensity-score matching should have prevented any major selection bias.

The results show that no differences could be determined in the incidence of MACCE between the two study groups over a period of 14 years or more (Figure 1, Figure 2 and Figure 3). On one side, this indirectly demonstrates that the presumable patency rate of device-dependent anastomoses should be similar to that of hand-sewn ones performed during double clamping CABG, since recurrence of angina or need for further treatment were substantially the same in both groups of patients.

On the other hand, data obtained from our study support the fact that if the surgical team is adequately trained in OPCAB, long-term results can be expected to be superimposable, whatever the technique is chosen.

The limitations in the use of the device have already been discussed in our previous studies. Besides that, the present study has some relevant limitations to be acknowledged: first, a direct evaluation of the actual patency rate was not obtained. Thus our conclusions are to be considered only reasonably certain; second, the study groups are numerically small due to the necessity of eliminating all cases performed by other surgeons with a lower level of experience in off-pump procedures. Subsequently, existing differences, although probably not clinically relevant, might not be elicited due to insufficient statistical power.

## 5. Conclusions

Our single-center study analysis suggests that clampless OPCAB with PAS-port assisted proximal anastomoses proved to be not inferior to double-clamping on-pump CABG in the long-term period, provided that involved surgeons are familiar with both techniques. These conclusions are supported by a large and long-term follow-up and the elimination of potential bias, i.e., by means of the propensity score matching.

## Figures and Tables

**Figure 1 ijerph-19-00275-f001:**
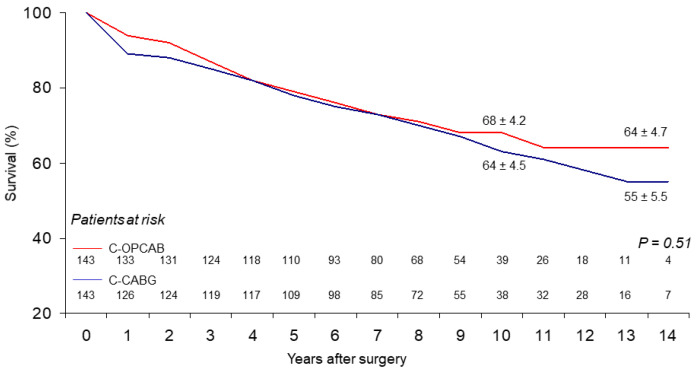
Long-term survival.

**Figure 2 ijerph-19-00275-f002:**
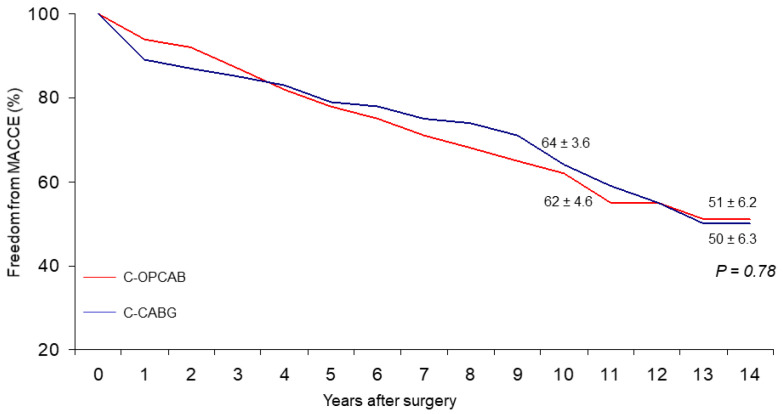
Freedom from MACCEs (MACCEs = Major Adverse Cardiac and Cerebrovascular Events).

**Figure 3 ijerph-19-00275-f003:**
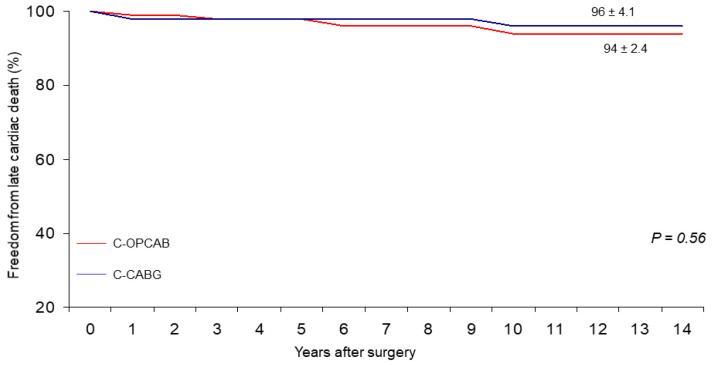
Freedom from late cardiac death.

**Table 1 ijerph-19-00275-t001:** Patient population’s characteristics before and after matching.

Variables	Overall Population (*n* = 366)			Propensity Score-Matched (*n* = 286)		
	C-OPCAB (*n* = 223)	C-CABG (*n* = 143)	*p*-Value	C-OPCAB (*n* = 143)	C-CABG (*n* = 143)	*p*-Value
Age, years	67.9 ± 9.7	67.4 ± 9.4	0.90	67.4 ± 9.4	67.4 ± 9.4	0.86
Female gender (%)	16.2	16.1	0.97	16.1	16.1	1
BMI (mean v. Kg/m^2^)	27.2 ± 3.7	27.5 ± 3.5	0.33	27.2 ± 3.7	27.5 ± 3.5	0.36
Hypertension (%)	85.1	89.5	0.23	88.1	89.5	0.71
Smoking habit (%)	28.8	31.5	0.59	29.4	31.5	0.71
Diabetes mellitus (%)	33.8	44.8	0.03	41.3	44.8	0.53
LVEF (mean v.)	0.51 ± 0.85	0.50 ± 0.99	0.03	0.52 ± 0.89	0.51 ± 0.99	0.46
Recent MI (%)	37.4	35	0.64	35.7	35	0.90
Previous PCI (%)	18	9.8	0.03	9.1	9.8	0.81
Renal failure (%) *	5.4	2.8	0.23	2.8	2.8	1
Previous CA stent (%)	2.7	2.8	0.96	2.1	2.8	0.65
CA stenosis 50–70% (%)	16.7	25.9	0.03	22.4	25.9	0.43
Stroke/TIA (%)	10.8	10.5	0.92	11.2	10.5	0.85
LMD ≥ 50% (%)	37.8	31.5	0.21	30.1	31.5	0.70
PVD (%)	27.5	39.9	0.01	35.7	39.9	0.40
Urgency (%)	30.2	28.7	0.76	31.5	28.7	0.60

BMI, body mass index; LVEF, left ventricular ejection fraction; MI, myocardial infarction; PCI, percutaneous coronary intervention; CA, carotid artery; TIA, transient ischemic attack; LMD, left main disease; PVD, peripheral vascular disease. * preoperative glomerular filtration less than 50 mL/min.

**Table 2 ijerph-19-00275-t002:** Actual rates of adverse events observed during follow-up, not including in-hospital results.

Adverse Events	C-OPCAB (*n* = 141)	C-CABG (*n* = 136)	*p*-Value
Death for any causes, *n* (%)	40 (28.4)	40 (29.4)	NS
New MI, *n* (%)	4 (2.9)	7 (5.0)	NS
New revascularization, *n* (%)	4 (2.9)	9 (6.4)	NS
Stroke, *n* (%)	1 (0.7)	0	NS
Pacemaker/ICD implantation, *n* (%)	3 (2.1)	10 (7.4)	NS

MI, myocardial infarction. NS = not significant.

**Table 3 ijerph-19-00275-t003:** Multivariable predictors of long-term survival and cardiac death.

**Survival**			
**Covariates**	**HR**	**95% CI**	***p*-Value**
Age at operation time (70.3 vs. 65.7 yrs)	4.5	1.041–1.109	<0.0001
Preoperative LVEF (0.50 vs. 0.52)	−2.4	0.940–0.993	0.015
**Late cardiac death**			
**Covariate**			
Preoperative LVEF (0.49 vs. 0.52)	−4.1	0.887–0.959	<0.0001

LVEF, left ventricular ejection fraction.

## Data Availability

Archived datasets analyzed at the Database of the Tor Vergata University Polyclinic.

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
