# Peer review of "Long-Term Follow-Up of Device-Assisted Clampless Off-Pump Coronary Artery Bypass Grafting Compared with Conventional On-Pump Technique"

_ijerph, 2021, doi:10.3390/ijerph19010275_

Round 1

Reviewer 1 Report

I have any suggestions for authors.

Author Response

  • we thank very much the Reviewer for his positive evaluation of the manuscript.

Reviewer 2 Report

Manuscript ‚Long-term follow-up of device-assisted clampless off-pump coronary artery bypass grafting compared with conventional on-pump technique’.

I thank for the opportunity to review the interesting manuscript, I have some questions and comments.

  1. Line 48, please add a reference.
  2. Lines 56 – 58. I do not understand what the authors want to express, patients were submitted by a surgeon?
  3. Table 1, head lines are probably shifted accidentally.
  4. Lines 151 to 161. A mortality rate of 4.9% in CABG is relatively high, even considering an urgency rate of 30%. Do the authors have other explanations?
  5. The same questions arise concerning a 9% rate of neurological complications.
  6. Lines 224 to 228. The authors stress the importance of complete revascularisation. The study cited includes a large number of OBCAB patients who underwent 4-7 distal anastomoses. In contrast, the mean number of anastomoses was 2.9 in the on-pump and even only 2.6 in the clampless off-pump group. Can the authors please comment on completeness of revascularisation in their patients.

Author Response

we thank the Reviewer 2 for his suggestions and comments to implement the manuscript; a) in the line 48 of the Introduction section: the reference has been added ([6,7]); b) in the lines 56-58 of the Materials and Methods section we have better explained the concept: the patients were operated on by a single expert surgeon; c) the table 1 has been checked; d) as now reported in the Results section, in the lines 156-157, we have explained the reasons of the mortality rate: 4 pts out of 7 deaths presented preoperatively an EuroSCORE II greater than 8%; e) the overall incidence of neurological injury was 2.1% (n = 3) in C-OPCAB group and 9.1% (n = 11) in C-CABG group: the higher incidence of neurological injury observed in C-CABG group was due primarily to a greater number of reversible neurological events, like transient ischemia or delirium (6.3% or 9 patients vs 2.1% or 3 patients), and the incidence of permanent neurologic injury (stroke) in the C-CABG group was 1.4% or 2 patients. We added these data at the lines 167-168; f) the rate of the complete revascularization has been reported in the Results section in the lines 152-154; of note, in the ROOBY trial the mean number per-patient was 2.9 and 3.0 following off-pump and conventional CABG, respectively. 

Reviewer 3 Report

Comments to the authors

Congratulate to author hard work. Several issues to ask

  • The clampless OPCAB with the proximal SVG anastomostic device theoretically is beneficial on the post-operative neurological complications. However, I do not see the results from the table 2 ( in table 2, it seem to me the neurological complications is after long-term follow up, it seemed to be more directly related to patient underlying comorbidities). Please explain
  • In the discussion, the author mentioned the long term survival is closely related to (1) complete revascularization (2) the quality of anastomosis. However, I did not see any clinical data in the result, comparing the quality of distal coronary artery between the clampless OPCAB and double clamping CABG
  • In Table 3: why the categorical variables, like age at operation time, pre-operative LVEF, are not divided by two separate variables ( for example, over 65 or less than 65, LVER over 50 or less than 50), instead, there is a gap in between, like (70.3 vs 65.7, then how about patients age in between)?
  • What is the statistical power to conclude the supplement survival curve?

Author Response

  • Reviewer 3: we thank very much the Reviewer for his positive evaluation of the manuscript; a) in Table 2 we have reported the actual incidence of late complications explaining that in-hospital results were not included; b) in this clinical study we have not evaluated the quality of the distal anastomoses by means with radiological examination (i.e., coronary CT data) ; c) in Table 3 we have reported the mean value of the age and of the LVEF as continuous covariates: we found that age greater than 70 years at operation and preoperative LVEF less than 0.50 were predictors of impaired late outcomes. In the text we have added the survival rate stratified by preoperative LVEF </= 0.35 in the lines 186-188.

Round 2

Reviewer 2 Report

I thank the authors for the responses to my questions and comments.